# Bis-Indole Derivatives as Dual Nuclear Receptor 4A1 (NR4A1) and NR4A2 Ligands

**DOI:** 10.3390/biom14030284

**Published:** 2024-02-27

**Authors:** Srijana Upadhyay, Amanuel Esayas Hailemariam, Fuada Mariyam, Zahin Hafiz, Gregory Martin, Jainish Kothari, Evan Farkas, Gargi Sivaram, Logan Bell, Ronald Tjalkens, Stephen Safe

**Affiliations:** 1Department of Veterinary Physiology, Texas A&M University, College Station, TX 77843, USA; supadhyay@cvm.tamu.edu (S.U.); ahailemariam@tamu.edu (A.E.H.); fuada@tamu.edu (F.M.); gmartin@cvm.tamu.edu (G.M.); farkasea@tamu.edu (E.F.); 2Department of Chemical Engineering, Texas A&M University, College Station, TX 77843, USA; zahin.hafiz22@tamu.edu (Z.H.); jainish.kothari@tamu.edu (J.K.); 3Department of Biochemistry and Biophysics, Texas A&M University, College Station, TX 77843, USA; gs715@tamu.edu; 4Department of Chemistry, University of La Verne, La Verne, CA 91750, USA; logan.bell@laverne.edu; 5Department of Environmental and Radiological Health Sciences, College of Veterinary Medicine and Biomedical Sciences, Colorado State University, Fort Collins, CO 80526, USA; ron.tjalkens@colostate.edu

**Keywords:** NR4A1, NR4A2, bis-indole, dual binding

## Abstract

Bis-indole derived compounds such as 1,1-bis(3′-indolyl)-1-(3,5-disubstitutedphenyl) methane (DIM-3,5) and the corresponding 4-hydroxyl analogs (DIM8-3,5) are NR4A1 ligands that act as inverse NR4A1 agonists and are potent inhibitors of tumor growth. The high potency of several DIM-3,5 analogs (IC_50_ < 1 mg/kg/day), coupled with the >60% similarity of the ligand-binding domains (LBDs) of NR4A1 and NR4A2 and the pro-oncogenic activities of both receptors lead us to hypothesize that these compounds may act as dual NR4A1 and NR4A2 ligands. Using a fluorescence binding assay, it was shown that 22 synthetic DIM8-3,5 and DIM-3,5 analogs bound the LBD of NR4A1 and NR4A2 with most K_D_ values in the low µM range. Moreover, the DIM-3,5 and DIM8-3,5 analogs also decreased NR4A1- and NR4A2-dependent transactivation in U87G glioblastoma cells transfected with GAL4-NR4A1 or GAL4-NR4A2 chimeras and a UAS-luciferase reporter gene construct. The DIM-3,5 and DIM8-3,5 analogs were cytotoxic to U87 glioblastoma and RKO colon cancer cells and the DIM-3,5 compounds were more cytotoxic than the DIM8-3,5 compounds. These studies show that both DIM-3,5 and DIM8-3,5 compounds previously identified as NR4A1 ligands bind both NR4A1 and NR4A2 and are dual NR4A1/2 ligands.

## 1. Introduction

The nuclear receptor (NR) superfamily of 48 receptors has been subdivided into several groups which include the endocrine receptors which bind vitamins and steroidal hormones, and adopted orphan receptors which bind small molecules including steroidal and lipid molecules and orphan receptors [1]. The endogenous ligands for orphan receptors have not yet been identified; however, the orphan receptors bind structurally diverse synthetic compounds, natural products, prostaglandins, and lipids. NRs share a common structural organization containing an N-terminal activation function 1 (AF1) domain adjacent to a DNA-binding domain (DBD) and a hinge region. The C-terminal AF2 domain of NRs contains a ligand-binding domain (LBD) which binds both endogenous and synthetic ligands. Ligand binding induces the interaction of the zinc finger DBD with cognate cis-acting response elements which are specific for each NR [1,2,3]. The activation of NRs by ligands or other factors initiates the transcriptional activation or inactivation of downstream genes and pathways that play critical roles in maintaining cellular homeostasis and in pathophysiology [3]. The expression and involvement of NRs in multiple diseases has resulted in their being a major target for drug development [4].

The nerve growth factor B (NGFB) subfamily of orphan receptors contains three members, namely NR4A1 (Nur77, TR3), NR4A2 (Nurr1), and NR4A3 (Nor1), and these receptors were initially identified as early intermediate response genes induced by the nerve growth factor in PC12 cells [5,6,7]. The three receptors have a common modular structure similar to NRs and their structural differences are primarily manifested in their respective N-terminal AF1 domain which exhibits only <20–30% homology. In contrast, the C-terminal ligand-binding domains (LBDs) of NR4A1, NR4A2, and NR4A3 exhibit approximately <60–65% homology [8]. Based on the results obtained with knockout mouse models and other studies, it is clear that NR4A members play significant roles in maintaining cellular homeostasis and in pathophysiology and they exhibit multiple individual activities and functions with only some overlap. For example, NR4A1^−/−^ mice are viable [9,10,11], NR4A2^−/−^ mice exhibit neuronal dysfunction and die within a few days after birth [12,13], and NR4A3^−/−^ mice generated in two laboratories were either embryonic lethal or viable but accompanied by inner ear defects [14,15]. Individually, both NR4A1 and NR4A2 exhibit pro-oncogenic activity in many solid tumors, whereas in many blood-derived cancers, NR4A1 and NR4A3 are classified as tumor suppressors based on the rapid development of acute myeloid leukemia observed in double knockout NR4A1^−/−^:NR4A3^−/−^ mice [16,17].

Early studies on the structure of NR4A2 suggested that because the LBD contained bulky amino acid substituents, NR4A members acted as ligand-independent transcription factors [18,19,20]. Subsequent studies have identified and characterized structurally diverse compounds that bind NR4A1, NR4A2, and NR4A3, and the functional effects of ligands that bind the former two receptors have been extensively investigated [21,22,23,24,25]. For example, the natural product cytosporone b (Csnb) and related compounds bind NR4A1, and the different analogs exhibit structure-dependent anticancer activity, variable effects on serum glucose levels, and the inhibition of drug-induced inflammation, fibrosis, and sepsis [26,27,28,29,30,31]. Studies in this laboratory investigated the activities of 1,1-bis(3′-indolyl)methane, a metabolite of the phytochemical indole-3-carbinol as a ligand for the aryl hydrocarbon receptor (AhR) and demonstrated that the addition of a substituted phenyl ring (CDIM) resulted in compounds that bound NR4A1 [25,32,33]. Several of these CDIM compounds including 1,1-bis(3′-indolyl)-1-(3,5-disubstituted phenyl) methane (DIM-3,5) analogs inhibited mammary tumor growth in an orthotopic athymic nude mouse model and IC_50_ values were <1 mg/kg/day [34]. The potency of the DIM-3,5 and the corresponding DIM8-3,5 analogs [35] which also contain a 4-hydroxyl group on the phenyl ring suggested that these compounds may not only act as inverse NR4A1 agonists, but also inhibit or inactivate other pro-oncogenic pathways. One possible additional target for these compounds is NR4A2 since NR4A2 is also pro-oncogenic in solid tumors and there is prior evidence that some bis-indole-derived compounds bind and inactivate NR4A2 [33,36]. Moreover, since there is a >60% similarity between the LBDs of NR4A1 and NR4A2, we hypothesize that DIM-3,5 and DIM8-3,5 compounds (Figure 1) are dual receptor ligands that bind both NR4A1 and NR4A2, and data confirming this hypothesis are reported in this manuscript.

## 2. Materials and Methods

Results of recent studies showed 1,1-bis(3′-indolyl)-1-(3,5-disubstituted-4-hydroxyphenyl) methane (DIM8-3,5) and 1,1-bis(3′-indolyl)-1-(3,5-disubstituted phenyl) methane (DIM-3,5) compounds were NR4A1 ligands that were potent inhibitors of tumor growth in orthotopic mouse xenograft model [34,35]. The following compounds (Figure 1) were synthesized for this study: DIM8-3,5 compounds: DIM8-3,5-Cl2, DIM8-3,5-Br_2_, DIM8-3,5-(Bu)_2_, DIM8-3,5-(CH3)_2_, DIM8-3,5-(OCH_3_)_2_, DIM8-3-Cl-5-F, DIM8-3-Br-5-OCH_3_, and DIM8-3-Cl-5-OCH_3_; and DIM-3,5-compounds: DIM-3,5-Cl_2_, DIM-3,5-Br_2_, DIM-3,5-F_2_, DIM-3,5-(CH_3_)_2_, DIM-3,5-(OCH_3_)_2_, DIM-3-Br-5-CF_3_, DIM-3-CF-5-CF_3_, DIM-3-F-5-CF_3_, DIM-3-Cl-5-OCF_3_, DIM-3-Br-5-OCF_3_, DIM-3-Br-5-OCH_3_, and DIM-3-Cl-5-OCF_3_. These compounds were synthesized by condensation of indole and a corresponding substituted aldehyde for 48 h at 90 °C in an aqueous acetic acid solution [water/acetic acid (1/0.1)] as described; the resulting condensation products were crystallized from petroleum spirit: benzene and were >97% pure [34,35]. The one-step synthesis of these compounds gave, primarily, a single condensation product and overall yields after crystallization varied from 85–95%. The substituted benzaldehydes used in the synthesis of DIM8-3,5 and DIM-3,5 compounds are listed in the order indicated above: DIM8-3,5 benzaldehydes: 3,5-dichloro-4-hydroxybenzaldehyde, 3,5-dibromo-4-hydroxy-benzaldehyde, 3,5-di-(t-butyl)-4-hydroxybenzaldehyde, 3,5-dimethyl-4-hydroxybenzaldehyde, 3,5-dimethoxy-4-hydroxybenzaldehyde, 3-chloro-5-fluoro-4-hydroxybenzaldehyde, 3-chloro-5-bromo-4-hydroxybenzaldehyde, 3-bromo-4-hydroxy-3-methoxybenzaldehyde, and 3-chloro-4-hydroxy-5-methoxybenzaldehyde; and DIM-3,5-benzaldehydes: 3,5-dichlorobenzaldehyde, 3,5-dibromobenzaldehyde, 3,5-difluorobenzaldehyde, 3,5-dimethylbenzaldehyde, 3,5-dimethoxybenzaldehyde, 3-bromo-5-trifluoromethylbenzaldehyde, 3-chloro-5-trifluoromethylbenzaldehyde, 3-fluoro-5-trifluoromethylbenzaldehyde, 3-chloro-5-trifluoromethoxybenzaldehyde, 3-fluoro-5-trifluoromethoxybenzaldehyde, 3-bromo-5-methoxybenzaldehyde, and 3-chloro-5-methoxybenzaldehyde. With the exception of 3,5-di t-butyl-4-hydroxybenzaldehyde, 3,5-difluorobenzaldehyde, 3,5-dimethoxybenzaldehyde, 3-bromo-5-trifluoro-methylbenzaldehyde, 3-chloro-5-trifluoromethylbenzaldehyde, 3-fluoro-5-trifluorobenzaldehyde, 3-chloro-3-fluoro-5-trifluoromethoxybenzaldehyde (all from Alfa Aesar, Ward Hill, MA, USA); 3-chloro-5-bromo-4-hydroxy benzaldehyde and 3-bromo-5-methoxy benzaldehyde (from AstaTech Inc., Bristol, PA, USA); and 3-bromo-4-hydroxy-5-methoxy benzaldehyde (J + H Chemical, Hangzhou, China), all the remaining substituted benzaldehydes and indoles were purchased from Sigma-Aldrich (St. Louis, MO, USA).

### 2.1. Chemical and Cell Culture Studies

The U87G glioblastoma and RKO colon cancer cell lines were obtained from ATCC (Manassas, VA, USA) and maintained in Dulbecco’s modified Eagle’s medium (DMEM). DMEM medium was supplemented with 10% fetal bovine serum (FBS; Sigma-Aldrich). Cells were maintained at 37 °C in the presence of 5% CO_2_. The solvent (DMSO) used in the experiments was always below 0.15% by volume in the medium. The DIM8-3,5 and DIM-3,5 analogs (Figure 1) were synthesized in our laboratory as previously described [34,35].

### 2.2. Cell Proliferation Assay

Cell proliferation was carried out using the XTT Cell Viability Kit (Cell Signaling Biotechnology, Danvers, MA, USA, Catalog # 9095). U87G and RKO cells were seeded at a density of 2 × 10^4^ cells/well with DMEM containing 10% FBS in a 96-well plate. Cells were incubated overnight and then treated with DMSO or different concentrations of CDIM (7.5 μM and 15 μM) in DMEM containing 2.5% charcoal-stripped FBS for 24 or 48 h. After 24 or 48 h of culture, XTT reagent with 1% electron coupling solution was added to each well and incubated for 4 h as per manufacturer’s instructions. The percentage of cell survival was calculated after measuring the absorbance at 450 nm wavelength in a plate reader. For each treatment, three independent experiments were carried out. Data were analyzed using GraphPad Prism for significance.

### 2.3. Luciferase Assay

Luciferase assays were carried out as previously described (34,35). The glioblastoma cell line U87G was used for transactivation assay. Cells were seeded on 24-well plate at a density of 3 × 10^4^/well in DMEM with 10% FBS. For NR4A1-dependent transactivation, cells were co-transfected with UASx5-Luc (300 ng), GAL4-NR4A1 (300 ng), and *β*-gal (50 ng) using GeneJuice as a transfection reagent (Sigma Aldrich, Catalog# 70967). Similarly, for NR4A2-dependent transactivation, cells were co-transfected with UASx5-Luc (300 ng), GAL4-NR4A2 (300 ng), and *β*-gal (50 ng) using GeneJuice. After 24 h, the medium was removed, and cells were treated with DMSO or different CDIM concentrations (7.5 μM and 15 μM) for 24 to 48 h. Cells were lysed using a freeze–thaw protocol and cell lysates were then processed to measure luciferase and *β*-gal activity. Galacto-Light Plus System (Applied Biosystems, Waltham, MA, USA) was used for the detection of *β-Glactosidase*. Luciferase activities were normalized against *β*-gal activity. Each experiment was carried out in triplicate, and results were analyzed using GraphPad Prism and expressed as means ± SE for each set of experiments.

### 2.4. Nuclear Magnetic Resonance Spectra Analysis Data

^1^HNMR Spectra were measured on a Bruker Avance III NEO Spectrometer (500 MHz) and data were analyzed on the Bruker TopSpin NMR Analysis Software (Version 3.7). Spectra of each sample in CDCl_3_ solvent are reported in ppm (s = singlet, d = doublet, t = triplet, q = quartet, m = multiplet). Integration coupling constant(s) in Hz was determined by using TMS as the internal standard at 0.00 ppm. The following are the results:

DIM-3,5-Cl_2_: δ 1.59 (s, 1H), 5.86 (s, 1H), 6.67 (s, 2H), 7.07 (t, 2H, *J* = 7.33), 7.23 (t, 4H, *J* = 7.78), 7.26 (s, 2H), 7.39 (dd, 4H, *J* = 2.4, 10.6), 7.95 (s, 2H)

DIM-3,5-Br_2_: δ 1.58 (s, 1H), 5.85 (s, 1H), 6.68 (s, 2H), 7.07 (t, 2H, *J* = 7.54), 7.22 (t, 2H, *J* = 7.42), 7.40 (m, 4H), 7.45, (d, 2H, *J* = 1.6), 7.99 (s, 2H)

DIM-3,5-F_2_: δ 1.59 (s, 1H), 5.89 (s, 1H), 6.69 (s, 2H), 6.9 (d, 2H, *J* = 6.9), 7.07 (t, 2H, *J* = 7.4), 7.23 (t, 2H, *J* = 7.7), 7.40 (t, 4H, *J* = 6.8), 7.95 (s, 2H)

DIM-3,5-(CH_3_)_2_: δ 2.28 (s, 6H), 5.84 (s, 1H), 6.66 (s, 2H), 6.89 (s, 1H), 7.01 (s, 1H), 7.04 (t, 2H, *J* = 7.54), 7.2 (t, 2H, *J* = 7.45), 7.4 (m, 5H), 7.84 (s, 2H)

DIM-3,5-(OCH_3_)_2_: δ 1.58 (s, 1H), 3.74 (s, 6H), 5.84 (s, 1H), 6.37 (s, 1H), 6.57 (s, 1H), 6.69 (s, 2H), 7.04 (t, 2H, *J* = 7.48), 7.19 (t, 2H, t = 7.65), 7.39 (s, 8H), 7.91 (s, 2H)

DIM-3-Br-5-CF_3_: δ 5.94 (s, 1H), 6.65 (s, 2H), 7.07 (t, 2H, *J* = 7.69), 7.24 (t, *J* = 2H, *J* = 7.69), 7.39 (t, 4H, *J* = 7.74), 7.61 (s, 1H), 7.67 (d, 2H, *J* = 8.9), 7.98 (s, 2H)

DIM-3-F-5-CF_3_: δ 1.63 (s, 1H), 5.97 (s, 1H), 6.8 (s, 2H), 7.06 (t, 2H, *J* = 7.4), 7.22 (m, 4H), 7.39 (t, 4H, *J* = 8.4), 7.5 (s, 1H), 7.99 (s, 2H)

DIM-3-F-5-OCF_3_: δ 1.6 (s, 1H), 5.92 (s, 1H), 6.7 (s, 2H), 6.86 (d, 1H, *J* = 9.15), 7.07 (m, 3H), 7.23 (t, 2H, *J* = 7.5), 7.40 (m, 5H), 7.98 (s, 1H)

DIM-3-Cl-5-OCH_3_: δ 1.58 (s, 1H), 3.74 (s, 3H), 5.84 (s, 1H), 6.70 (s, 2H), 6.77 (s, 1H), 6.84 (s, 1H), 6.96 (s, 1H), 7.04 (t, 2H, *J* = 7.48), 7.20 (t, 2H, *J* = 7.5), 7.39 (m, 4H), 7.96 (s, 1H)

DIM8-3,5-Cl_2_: δ 1.6 (s, 1H), 5.81 (s, 1H), 6.68 (s, 2H), 7.06 (t, 2H, *J* = 7.54), 7.25 (m, 4H), 7.40 (m, 6H), 7.98 (s, 2H)

DIM8-3,5-Br_2_: δ 1.59 (s, 1H), 3.73 (s, 4H), 5.81 (s, 1H), 6.55 (s, 1H), 7.06 (t, 1H, *J* = 7.5), 7.25 (q, 2H, *J* = 6.5), 7.35 (d, 1H, *J* = 8.29), 7.43 (m, 3H)

DIM8-3,5-(tBu)_2_: δ 1.38 (s, 18H), 1.41 (s, 2H), 1.58 (s, 2H), 5.04 (s, 1H), 5.81 (s, 1H), 6.73 (s, 2H), 7.01 (t, 2H, *J* = 7.25), 7.18 (m, 4H), 7.39 (m, 5H), 7.91 (s, 2H)

DIM8-3,5-(CH_3_)_2_: δ 1.59 (s, 1H), 2.20 (s, 6H), 4.51 (s, 1H), 5.79 (s, 1H), 6.56 (s, 2H), 6.98 (s, 2H), 7.04 (t, 2H, *J* = 7.5), 7.19 (t, 2H, *J* = 7.65), 7.41 (m, 8H), 7.85 (s, 2H)

DIM8-3,5-(OCH_3_)_2_: δ 1.59 (s, 1H), 3.76 (s, 6H), 5.83 (s, 1H), 6.62 (s, 2H), 6.68 (bs, 2H), 7.03 (t, 2H, *J* = 7.42), 7.19 (t, 2H, *J* = 7.62), 7.37 (d, 2H, *J* = 8.17), 7.42 (d, 2H, *J* = 7.88), 7.95 (s, 2H)

DIM8-3-Cl-5-F: δ 1.6 (s, 1H), 5.39 (s, 1H), 5.82 (s, 1H), 6.64 (s, 2H), 7.01 (dd, 1H, *J* = 1.95, 9.15), 7.06 (t, 2H, *J* = 8.02), 7.16 (bs, 1H), 7.23 (t, 2H, *J* = 7.37), 7.39 (m, 7H), 7.97 (s, 2H)

DIM8-3-Cl-5-Br: δ 1.60 (s, 1H), 5.79 (s, 2H), 6.67 (s, 2H), 7.07 (t, 2H, *J* = 7.9), 7.29 (t, 2H, *J* = 7.28), 7.29 (m, 1H), 7.39 (m, 8H), 7.95 (s, 2H)

DIM8-3-Br-5-OCH_3_: δ 1.58 (s, 3H), 3.79 (s, 3H), 5.82 (s, 2H), 6.69 (s, 2H), 6.87 (s, 1H), 7.05 (m, 3H), 7.21 (t, 2H, *J* = 7.56), 7.29 (s, 1H), 7.40 (dd, 4H, *J* = 2.7, 8.10), 7.96 (s, 2H)

DIM8-3-Cl-5-OCH_3_: δ 1.58 (s, 3H), 3.80 (s, 3H), 5.73 (s, 1H), 5.79 (s, 1H), 6.69 (s, 2H), 6.84 (s, 1H), 6.92 (s, 1H), 7.04 (t, 2H, *J* = 7.46), 7.21 (t, 2H, *J* = 7.64), 7.28 (s, 1H), 7.40 (m, 4H), 7.96 (s, 1H)

### 2.5. Direct Binding Assay [34,35]

Quenching of NR4A1 and NR4A2 tryptophan fluorescence by direct ligand binding was conducted using the Varian Cary Eclipse Fluorescence Spectrophotometer at 25 °C. The ligand-binding domain (LBD) of NR4A1/A2 (0.5 μM) in phosphate buffered saline (pH 7.4) was incubated with different concentrations of ligands. Fluorescence was obtained using excitation and emission wavelengths of 285 nm (excitation slit width = 5 nm) and 300–420 nm (emission slit width = 5 nm), respectively. Data analysis was performed using Sigma Plot. Ligand-binding K_D_ and R^2^ values were determined by measuring concentration-dependent NR4A1/A2 tryptophan fluorescence intensity at an emission wavelength of 330 nm.

### 2.6. Computation-Based Molecular Modeling Studies

Molecular modeling studies were conducted using Maestro (Schrödinger Release 2021-3, Schrödinger, LLC, New York, NY, USA, 2021). The version of Maestro used for these studies was licensed to the Laboratory for Molecular Simulation (LMS), a Texas A&M University core-user facility for molecular modeling and is associated with the Texas A&M University High-Performance Research Computing (HPRC) facility (College Station, TX 77843, USA). All Maestro-associated applications were accessed via the graphical user interface (GUI)’s VNC interactive application through the HPRC OnDemand portal. The crystal structure coordinates for the human orphan nuclear receptor NR4A1 and NR4A2 LBD [30,36] were downloaded from the Protein Data Bank (https://www.rcsb.org; PDB ID 3V3Q,5Y41). The human NR4A1 LBD and NR4A2 LBD crystal structures were prepared for ligand docking utilizing the Maestro Protein Preparation Wizard; restrained minimization of the protein structure was performed utilizing the OPLS4 force field. The three-dimensional structure of each ligand was prepared for docking utilizing the Maestro LigPrep. Maestro Glide [37,38,39] was utilized with the default settings to dock each ligand to each protein, predict the lowest energy ligand-binding orientation, and calculate the predicted binding energy in units of kcal/mol.

### 2.7. Statistical Analysis

One-way ANOVA was used to measure significance between control and multiple treatment groups for cell viability results. In order to confirm the reproducibility of the data, the in vitro cell culture experiments were performed at least three independent times, and results were expressed as means ± SD. *p*-values less than 0.05 were statistically significant.

## 3. Results

This study investigates the dual receptor-binding activity of a series of DIM-3,5 and DIM8-3,5 compounds (Figure 1) that exhibit potent in vivo anticancer activity [34,35]. Table 1 summarizes the direct binding of 22 analogs to the LBD of NR4A1 and NR4A2 using a fluorescence quenching assay as previously described [34,35].

The results show that each of these compounds bind both NR4A1 and NR4A2, with respective K_D_ values for binding NR4A1 ranging from 1.3–133 µM and for binding NR4A2 the range of the K_D_ values was 2.2–79 µM. Among the DIM-3,5 and DIM8-3,5 analogs, the compound with the highest K_D_ values for binding NR4A1 and NR4A2 using the fluorescence assay was DIM-3,5-(CH_3_)_2_. The K_D_ values were lower for the corresponding DIM8-3,5-(CH_3_)_2_ compounds; however, the K_D_s for this ligand were among the highest for the DIM8-3,5 analogs. For the remaining compounds, there was not an apparent structure–binding relationship for K_D_ values associated with their binding to NR4A1 and NR4A2 or for their corresponding K_D_(NR4A1)/K_D_(NR4A2) ratios. Figure 2A,B illustrates the direct binding of DIM-3,5-CI_2_ to the LBD of NR4A1 and NR4A2 with K_D_ values of 7.7 and 12.0 µM, respectively. Figure 2C,D illustrates the interaction of DIM-3,5-CI_2_ with various amino acid side-chains in the LBD. Figure 3 illustrates the comparable interactions of DIM-3,5-Br_2_ with NR4A1 and NR4A2 where the K_D_ values were 6.5 and 12.2 µM, respectively, and interactions with amino acid side-chains in the LBD are indicated (Figure 3C,D). The docking scores for DIM-3,5-CI_2_ and DIM-3,5-Br_2_ were −5.57 and −4.37, and 2.76 kcal/mol for binding NR4A2. The modeling results and docking scores for all compounds are summarized in Table 2.

Maestro/Schrodinger software (Version 2021-3) was used in molecular modeling studies [37,38,39] to investigate interactions between the DIM-3,5 and DIM8-3,5 analogs with the TMY301 and TMY302 binding sites previously identified in the LBD of NR4A1 [29] (Appendix A). Among the DIM-3,5 analogs, only DIM-3,5-Br_2_, DIM-3-Br-5-CF_3_, and DIM-3-Cl-5-CF_3_ did not exhibit a lower docking score for the TMY301 site, whereas for the DIM8-3,5 analogs, all compounds except DIM-3,5-(tBu)_2_ exhibited a lower docking score for the TMY301 site in NR4A1. Figure 2 and Figure 3 illustrate the calculated interactions of DIM-3,5-CI_2_ and DIM-3-CI-5-CF_3_ with TMY301 which, like TMY302, is close to the surface of the ligand binding pocket. Amino acids involved in the ligand–LBD interactions are somewhat variable and Figure 4A,B illustrates the specific interactions between DIM8-3,5-(CH_3_)_2_ and DIM-3,5-Br_2_ which have large differences in docking scores of −6.04 and −4.00 kcal/mole, respectively. The results of the modeling studies show that both compounds interact with Glu114, Leu113, Ser110, Glu109, Ile260, Thr264, Pro266, Leu239, Thr236, Cys235, Arg232, and Arg184 and these are the only amino acids associated with DIM-3,5-Br_2_–LBD interactions. In contrast, DIM8-3,5-(CH_3_)_2_, with the more favorable docking score (−6.04 kcal/mol), also interacts with amino acids Ala111, Phe112, Leu231, and Leu228, and presumably these amino acids contribute to the lower docking score for this compound.

The results of the modeling studies comparing congeners with low (DIM-3-F-5-CF_3_; −5.96) and high (DIM8-3,5-(t-Bu)_2_; −2.44) docking scores representing favorable and less favorable interactions with NR4A2 [36], respectively, are illustrated in Figure 4C,D. Although both compounds induce interactions with nine amino acid side-chains in common in NR4A2 (Leu437, Ser437, Ser441, Glu445, Arg563, Cys566, Leu570, Thr595, and Pro597), the differences in their docking scores are associated with the unique interaction of the bound NR4A2 complex with amino acids in the LBD. For example, DIM-3-F-5-CF_3_ induced interactions with Ala442, Phe443, Leu444, Arg515, Thr567, Ile573, Leu591, Phe592, Leu596, and Phe598, whereas DIM8-3,5-(tBu)_2_ uniquely induced interactions with Leu559, Pro560, Thr513, Glu514, Arg515, and His516.

The effects of the DIM-3,5 and DIM8-3,5 analogs on transactivation were determined using U87G human glioblastoma cells transfected with GAL4-NR4A1 (LBD) or GAL4-NR4A2 (LBD) and the UAS-luc reporter gene, which contains five tandem GAL4-DNA binding sites. The U87G cell line expresses both NR4A1 and NR4A2 [36] and the results obtained after the treatment of the U87G cells with 7.5 and 15 µM of DIM-3,5 compounds for 24 h showed that all ligands decreased transactivation in cells transfected with GAL4-NR4A1 or GAL4-NR4A2 (Figure 5A,B). These results complement the binding of the DIM-3,5 analogs to NR4A1 and NR4A2 and further confirm that these compounds are dual receptor ligands. Preliminary studies with the series of DIM8-3,5 analogs showed minimal effects on the luciferase activity after treatment for 24 h. However, after incubation for 48 h, these compounds also decreased NR4A1- and NR4A2-dependent transactivation in U87G cells (Figure 5C,D). The potencies for most of the DIM8-3,5 analogs were less than those observed for the DIM-3,5-compounds and this correlated with their reported differences in in vivo potencies as inhibitors of mammary tumor growth in an orthotopic athymic nude mouse xenograft model [34,35].

In previous studies, we have shown that the CDIM compounds inhibit the growth of multiple cancer cell lines and, in breast cancer cells, the knockdown of NR4A1 abrogated the growth inhibitory effects of DIM-3-Cl-5-OCH_3_, DIM-3-Cl-5-OCF_3_, DIM-3-Cl-5-CF_3_, and DIM-3-Br-5-OCF_3_ [33,34,35]. The cytotoxicity of the DIM-3,5 and DIM8-3,5 analogs were further investigated in U87G and human RKO colon cancer cells. In U87G cells, treatment with 7.5 and 15 µM DIM-3,5 analogs for 24 h significantly decreased cell proliferation at both concentrations and similar results were observed for all analogs (Figure 6A). In contrast, using the same treatment protocol (i.e., 24 h incubation), the DIM8-3,5, in Figure 6B, ligands were much less potent than the DIM-3,5 compounds; however, after treatment for 48 h, both the 7.5 and 15 µM concentrations of DIM8-3,5 ligands inhibited growth, except for DIM8-3-Cl-5-Br (Figure 6C). This lack of potency may be due to the insolubility of DIM8-3-Cl-5-Br, since a fine white precipitate appeared after the addition of this compound into the medium. In contrast, RKO human colon cancer cell lines were more susceptible to the effects of 7.5 and 15 µM DIM-3,5 and DIM8-3,5 analogs since growth inhibition was observed for most compounds (exception: DIM8-3,5-Br_2_ at 24 h) after treatment for 24 h (Figure 7A,B).

Thus, both the DIM-3,5 and DIM8-3,5 ligands bind NR4A1 and NR4A2, activate NR4A1- and NR4A2-dependent transactivation, and inhibit the growth of U87G and RKO cells. In the transactivation and growth inhibition studies, the DIM-3,5 compounds were more potent than the DIM8-3,5 analogs, and this corresponded to the relative in vivo potencies of some of these analogs [34,35]. However, within both sets of ligands there was not an obvious structure–activity relationship with respect to receptor binding (two assays) transactivation or the inhibition of cell proliferation (two cell lines). In summary, this study describes a series of structurally related bis-indole compounds that bind both NR4A1 and NR4A2 and are dual receptor ligands that act as inverse NR4A1/2 agonists in cancer cell lines.

## 4. Discussion

The NR4A subfamily of NRs are potential drug targets for treating multiple diseases and this is due, in part, to their enhanced tissue/organ specific expression associated with cellular stress and inflammation [40,41,42,43,44]. Several studies have focused on the identification of ligands that bind NR4A1; however, their effects in various animal models are structure-dependent. For example, in mouse models of metabolic disease, the NR4A1 ligand, Csnb, elevated serum glucose levels [26], whereas structurally related ethyl-2-[2,3,4-trimethoxy-6-(1-octanoyl) phenyl acetate (TMPA) decreased blood glucose levels in diabetic mouse models [30]. Another Csnb analog, n-pentyl-2-(nonanoyl) phenyl acetate (PDNPA), inhibits NFkB-dependent transactivation in RAW264.7 cells by blocking interactions of NR4A1 with p38⍺ in a mouse model for sepsis [31]. In contrast, TMPA and another Csnb analog, 1-(3,4,5-trihydroxyphenyl) nonan-1-one (THPN), are inactive in this assay and Csnb has only minimal in vitro activity and no effect on sepsis in vivo [31]. PDPNA binds NR4A1, NR4A2, and NR4A3 but inhibits sepsis only through NR4A1. A recent comprehensive study on NR4A2 ligands reported that Csnb also bound NR4A2 using protein NMR structural footprinting, whereas TMPA did not bind NR4A2 [24]. These studies clearly demonstrate not only that the NR4A ligands are selective (tissue/response-specific) receptor modulators, but they also exhibit NR4A2-independent activities, and this has been observed for other NR4A ligands [24].

DIM-3,5 analogs were previously identified as NR4A1 ligands in both binding and transactivation assays and these compounds interact directly with the LBD of this receptor [34,35,45]. In contrast, 1,1-bis(3′-indolyl)-1-(4-chlorophenyl) methane (DIM12) and the 4-bromo analog did not bind NR4A1, and modeling studies showed that DIM12 preferentially bound the cofactor site in the LBD of NR4A2 [46]. Ongoing studies confirm that the NR4A2 activity of DIM12 is associated with interactions with the cofactor site and this compound does not interact with the ligand-binding pocket of NR4A2 or NR4A1. This study on structurally related DIM8-3,5 and DIM-3,5 analogs (Figure 1) was initiated because of their potent anticancer activity [34,35] and the fact that this enhanced activity could be due, in part, to the simultaneous inhibition of both pro-oncogenic NR4A1- and NR4A2-regulated genes/responses. This notion of dual receptor binding is also supported by the >60% identity of the respective LBDs of NR4A1 and NR4A2. Our results, using a direct binding (fluorescence) assay, showed that these compounds bound both NR4A1 and NR4A2 and act as dual receptor ligands, and they also inhibit both NR4A1- and NR4A2-dependent transactivation. Our modeling studies show a wide variation of docking scores and ligand interactions with side-chain amino acids of NR4A1 and NR4A2 (Figure 2, Figure 3 and Figure 4) and for NR4A1 there were some indications of specific LBD amino acid side-chain–ligand interactions that decreased the docking scores. However, the docking score values for individual compounds did not show any consistent correlations with the corresponding K_D_ values using the direct binding assay. We also confirmed the binding of DIM-3,5 and DIM8-3,5 compounds to NR4A1 and NR4A2 using an isothermal titration calorimetry assay; however, the results were variable due to technical problems, and this is currently being investigated.

Structure–activity studies over the various binding, transactivation, and cytotoxicity assays did not consistently identify individual three or five substituents that enhance activity since most compounds were active. Similar results were observed in testing four DIM-3,5 and 4 DIM8-3,5 analogs where their differences in potency as inhibitors of breast tumor growth in athymic nude mice were comparable within each subset of compounds. Their potency as inverse agonists are consistent with their inactivation of pro-oncogenic NR4A1 and NR4A2 in solid tumor-derived cell lines.

## 5. Conclusions

In summary, results of this study demonstrate that DIM-3,5 and DIM8-3,5 analogs bind both NR4A1 and NR4A2, activate NR4A1- and NR4A2-dependent transactivation, and, like other DIM analogs, these compounds are cytotoxic to colon and glioblastoma cells [32,36]. The individual series of DIM-3,5 and DIM8-3,5 compounds are substituted at the 3,5- and 3,4-hydroxy, five positions, respectively, differing only with respect to their three and five substituents, and their activities in terms of transactivation and cytotoxicity were similar. Although there were some binding-assay-specific differences in their K_D_ values and K_D_(NR4A1)/K_D_(NR4A2) ratios, each of the DIM-3,5 and DIM8-3,5 analogs bound both receptors and are dual receptor ligands that act as inverse agonists in cancer cell lines. These structural and functional similarities do not preclude a role for DIM-3,5 and DIM8-3,5 analogs acting as tissue/response-specific selective receptor modulators and this will be investigated in future studies. In addition, we will also investigate the applications of dual NR4A1/NR4A2 ligands on responses in which both receptors play similar roles, and this includes neuroinflammation and immune cell exhaustion [47,48,49,50], where it has recently been reported that DIM8-3-Cl-5-OCH_3_ enhances CD8^+^ T-cell/CD4^+^ T-cell ratios in tumor-infiltrating lymphocytes [32]. Moreover, we have also reported that this compound also inhibited tumor growth in a syngeneic mouse model of colon cancer using MC-38 cells as xenografts and the analysis of T cells in tumor-infiltrating lymphocytes showed that the dual NR4A1/2 ligands reversed T-cell exhaustion [51].

## Figures and Tables

**Figure 1 biomolecules-14-00284-f001:**
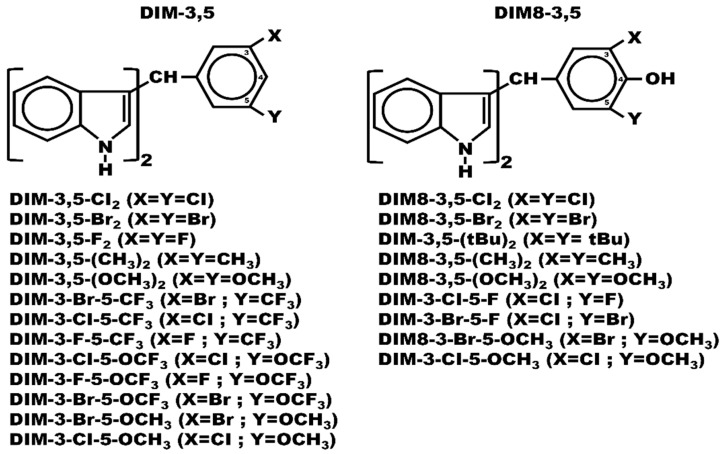
Structures of DIM-3,5 and Dim8-3,5 and their analogs used in this study.

**Figure 2 biomolecules-14-00284-f002:**
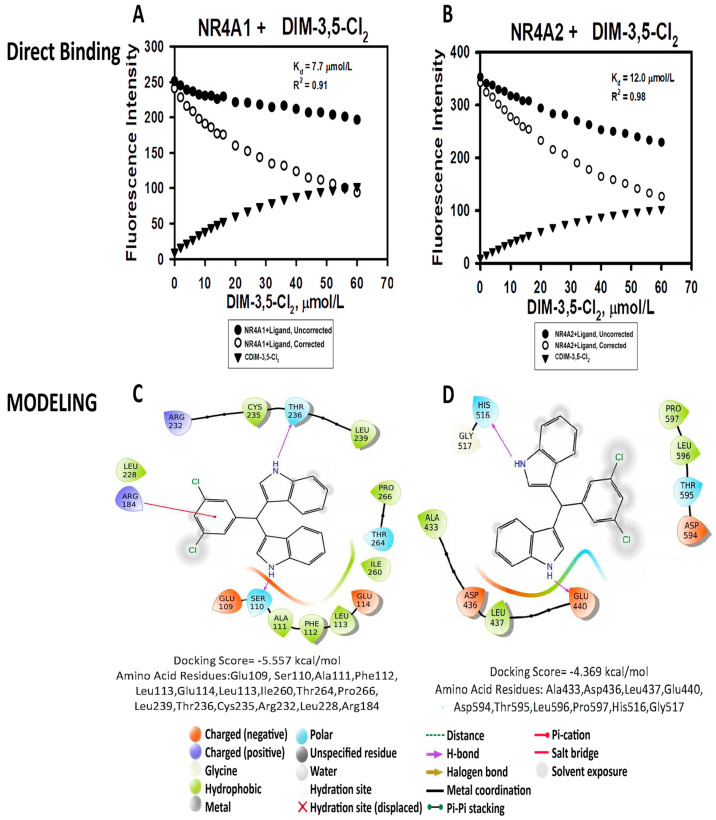
Binding of DIM-3,5-CI_2_ to NR4A1 and NR4A2: The LBD of NR4A1 (**A**) and NR4A2 (**B**) were incubated with DIM-3,5-CI_2_-binding curves for DIM-3,5-CI_2_ alone (
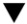
), uncorrected (
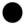
), and corrected (
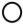
) (uncorrected-background ligand binding) were determined as outlined in the Methods. Modeling interactions of DIM-3,5-CI_2_ with LBD of NR4A1 (**C**) and NR4A2 (**D**) were determined as outlined in the Methods.

**Figure 3 biomolecules-14-00284-f003:**
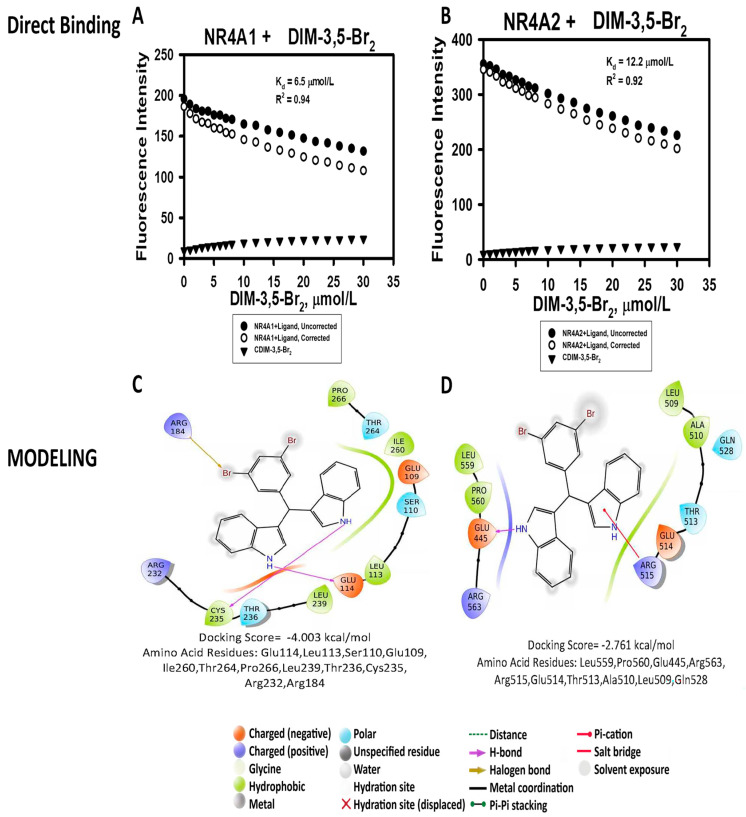
Binding of DIM-3,5-Br_2_ to NR4A1 and NR4A2: The LBD of NR4A1 (**A**) and NR4A2 (**B**) were incubated with DIM-3,5-Br_2_-binding curves for DIM-3,5-Br_2_ alone (
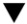
), uncorrected (
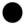
), and corrected (
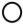
) (uncorrected-background ligand binding) were determined as outlined in the Methods. Modeling interactions of DIM-3,5-Br_2_ with LBD of NR4A1 (**C**) and NR4A2 (**D**) were determined as outlined in the Methods.

**Figure 4 biomolecules-14-00284-f004:**
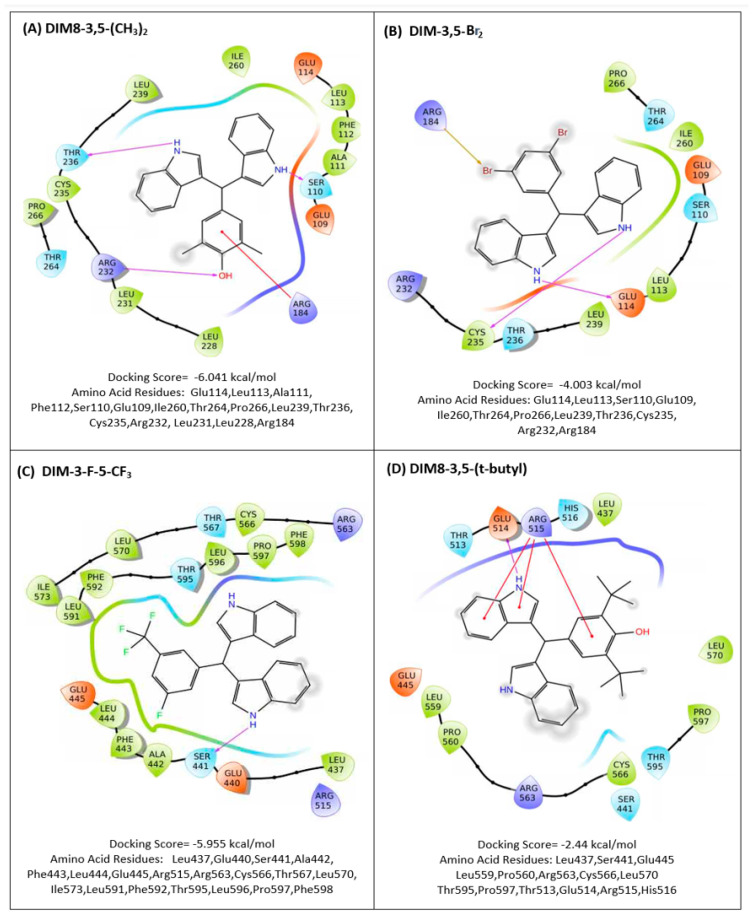
Modeling of DIM−3,5 and DIM8−3,5 interactions with NR4A1 and NR4A2, The interactions of DIM8−3,5−(CH_3_)_2_ (**A**) and DIM−3,5−Br (**B**) with amino acid side-chains of NR4A1 (LBD) and interactions of DIM−3−F−5−CF_3_ (**C**) and DIM8−3,5−(tBu)_2_ (**D**) with NR4A2 were determined using the Maestro/Schrodinger software [37,38,39], as outlined in the Methods.

**Figure 5 biomolecules-14-00284-f005:**
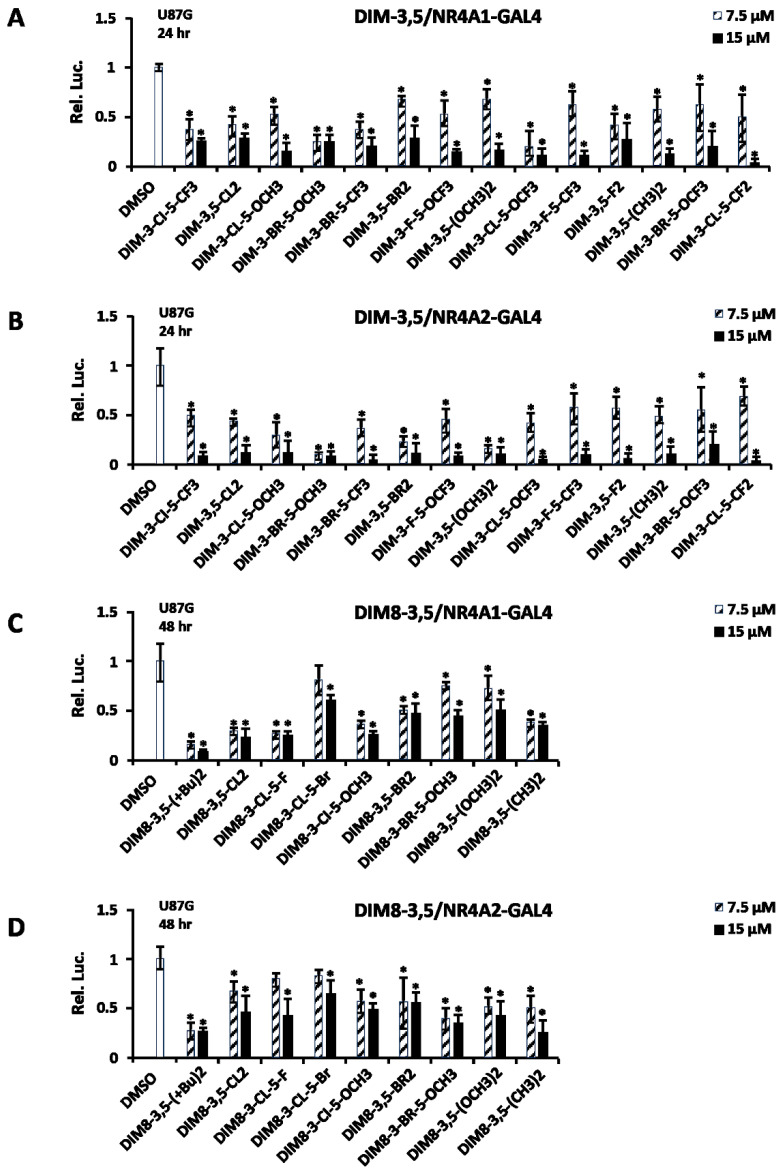
Activation of NR4A1 and NR4A2 by DIM-3,5 and DIM8-3,5 analogs. U87G cells were transfected with UAS-Luc and GAL4-NR4A1 (**A**) or GAL4-NR4A2 (**B**), treated with DIM-3,5 compounds (7.5 and 15 µM) and luciferase activity was determined as outlined in the Methods. The same protocol was used for activation of NR4A1- (**C**) and NR4A2-dependent (**D**) activities by DIM8-3,5 compounds. Results are expressed as means ± SD for at least three replicate determinations and significant (*p* < 0.05) changes compared to DMSO (control) are indicated (*).

**Figure 6 biomolecules-14-00284-f006:**
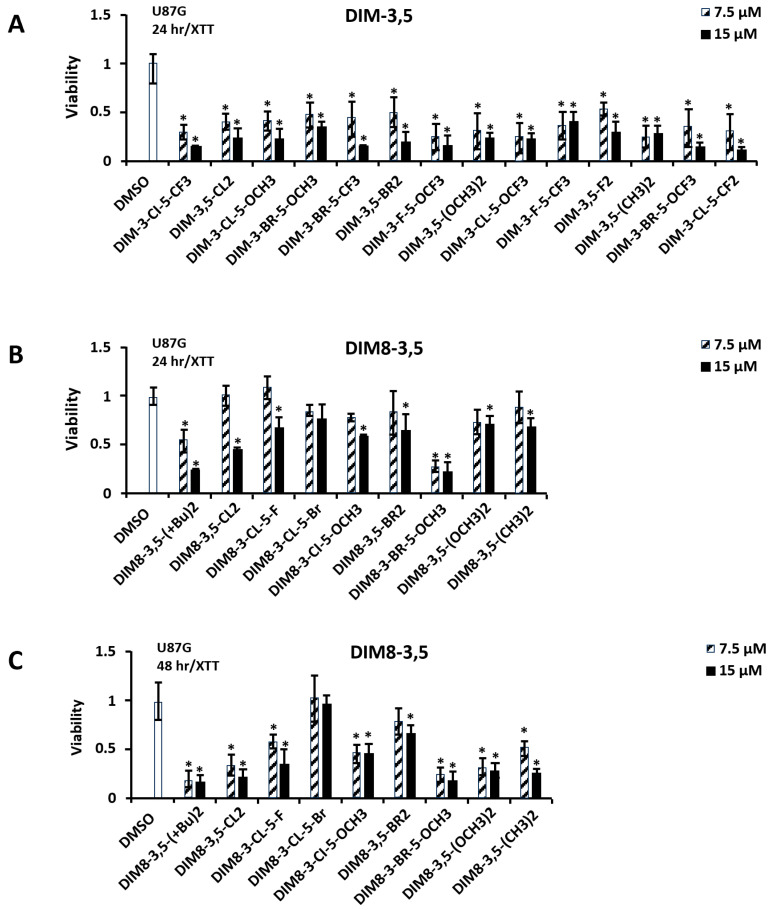
Cytotoxicity of DIM-3,5 and DIM8-3,5 analogs in U87G cells. Cells were treated with DIM-3,5 (**A**) or DIM8-3,5 (**B**) analogs for 24 h and also DIM8-3,5 analogs for 48 h (**C**), and activity in an XTT assay was determined as outlined in the Methods. Results are expressed as means ± SD for at least 3 replicates. Significant (*p* < 0.05) inhibition is indicated (*).

**Figure 7 biomolecules-14-00284-f007:**
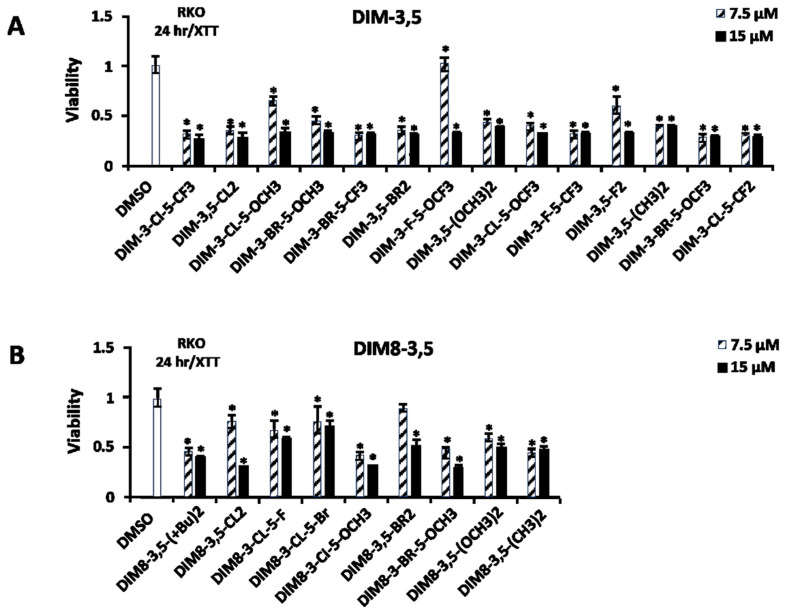
Cytotoxicity of DIM-3,5 and DIM8-3,5 analogs in RKO cells. Cells were treated with 7.5 or 15 µM of the test compounds for 24 h and activity in an XTT assay (**A**,**B**) was determined as outlined in the Methods. Results (**A**,**B**) are expressed as means ± SD for at least 3 replicate determinations and significant (*p* < 0.05) inhibition is indicated (*).

**Table 1 biomolecules-14-00284-t001:** Direct Binding of DIM-3,5- and DIM8-3,5-compounds to the LBD of NR4A1 and NR4A2.

Compound	K_D_ (NR4A1) (µM)	K_D_ (NR4A2) (µM)	Compound	K_D_ (NR4A1) (µM)	K_D_ (NR4A2) (µM)
DIM-3,5-Cl_2_	7.7	12.0	DIM8-3,5-Cl_2_	4.1	13.9
DIM-3,5-Br_2_	6.5	12.2	DIM8-3,5-Br_2_	1.3	7.4
DIM-3,5-F_2_	17.8	6.49	DIM8-3,5-(tBu)_2_	7.9	6.5
DIM-3,5-(CH_3_)_2_	133	79	DIM8-3,5-(CH_3_)_2_	24.5	10.7
DIM-3,5-(OCH_3_)_2_	15.8	9.97	DIM8-3,5-(OCH_3_)_2_	4.0	14.3
DIM-3-Br-5-CF_3_	4.8	4.9	DIM8-3-Cl-5-F	1.5	4.2
DIM-3-Cl-5-CF_3_	3.1	5.5	DIM8-3-Cl-5-Br	4.3	4.3
DIM-3-F-5-CF_3_	7.3	8.1	DIM8-3-Br-5-OCH_3_	6.7	7.3
DIM-3-Br-5-OCF_3_	2.0	4.5	DIM8-3-Cl-5-OCH_3_	6.6	2.2
DIM-3-Cl-5-OCF_3_	2.3	3.5			
DIM-3-F-5-OCF_3_	17.1	33.6			
DIM-3-Br-5-OCH_3_	1.3	3.5			
DIM-3-Cl-5-OCH_3_	60.3	5.2			

**Table 2 biomolecules-14-00284-t002:** Modeling of Interactions of DIM-3,5 and DIM8-3,5 with the LBD of NR4A1 and NR4A2.

Compound	Docking Scores NR4A1 TMY301/ TMY302	(kcal/mol) NR4A2	Compound	Docking SCORES NR4A1 TMY301/ TMY302	(kcal/mol) NR4A2
DIM-3,5-Cl_2_	−5.56/−5.03	−4.46	DIM8-3,5-Cl_2_	−4.84/−4.68	−3.63
DIM-3,5-Br_2_	−4.00/−4.44	−3.47	DIM8-3,5-Br_2_	−3.85/−4.04	−3.61
DIM-3,5-F_2_	−5.25/−4.40	−3.81	DIM8-3,5-(tBu)_2_	−5.14/−5.85	−2.44
DIM-3,5-(CH_3_)_2_	−5.12/−4.82	−4.32	DIM8-3,5-(CH_3_)_2_	−6.04/−4.45	−3.70
DIM-3,5-(OCH_3_)_2_	−5.98/−5.42	−2.93	DIM8-3,5-(OCH_3_)_2_	−5.96/−5.58	−5.30
DIM-3-Br-5-CF_3_	−4.06/−5.31	−3.69	DIM8-3-Cl-5-F	−5.68/−4.77	−2.74
DIM-3-Cl-5-CF_3_	−4.88/−5.52	−3.93	DIM8-3-Cl-5-Br	−4.49/−4.48	−4.06
DIM-3-F-5-CF_3_	−5.56/−4.52	−5.96	DIM8-3-Br-5-OCH_3_	−5.28/−4.91	−4.03
DIM-3-Br-5-OCF_3_	−5.08/−4.23	−2.93	DIM8-3-Cl-5-OCH_3_	−5.41/−4.97	−4.28
DIM-3-Cl-5-OCF_3_	−5.33/−4.57	−2.92			
DIM-3-F-5-OCF_3_	−4.78/−4.41	−5.34			
DIM-3-Br-5-OCH_3_	−5.34/−5.01	−3.69			
DIM-3-Cl-5-OCH_3_	−5.78/−5.29	−4.36			

## Data Availability

Data will be made available upon request.

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
