# Peer review of "Bis-Indole Derivatives as Dual Nuclear Receptor 4A1 (NR4A1) and NR4A2 Ligands"

_biomolecules, 2024, doi:10.3390/biom14030284_

Round 1

Reviewer 1 Report

Comments and Suggestions for Authors

The authors describe novel inverse agonists for NR4A1- the receptor critical in regulating tumor growth. In all the paper i well executed with standard biochemical and chemistry assays. I would ask the authors to submit original data files to an NIH-type repository and include that "link" in the manuscript. In addition, I would advise a modeling expert to review the model for docking the structures as provided in the supplementary data. The modeling parameters also need to be submitted to a repository. The remaining data are fairly straight forward; however, I have not verified the NMR and MS characteristics of the synthesized compounds. Here chemistry input would be important (e.g., purity and yields at each step). The authors must document the source of the cell line used and its validation. 

Author Response

We have included almost all of the results of this study and have now amplified the 2D presentation of the ligand-receptor interactions. If this supplemental data is not included in the manuscript we will submit it to a repository. The details on the synthetic procedures have now been amplified in the text and sources of the cell lines are now indicated.

Reviewer 2 Report

Comments and Suggestions for Authors

Article is in general good written. 

Please upload Figures 5, 6 and 7 in higher resolution minimum 600 dpi, also ad tables with values for those figures as well as a supplementary data.

Author Response

The resolution of the Figures (5-7) have been completely redone and are improved. We have also enlarged the supplementary data so that interactions of the ligand with side-chain amino acids of the LBD can be ascertained.

Reviewer 3 Report

Comments and Suggestions for Authors

The article "Bis-Indole Derivatives as Dual Nuclear Receptor 4A1 (NR4A1) and NR4A2 Ligands" presents a comprehensive study on bis-indole derived compounds, particularly focusing on DIM-35 and DIM8-35 analogs, as potential dual ligands for NR4A1 and NR4A2 nuclear receptors. The study outlines the synthesis of these compounds, their binding affinity to NR4A1 and NR4A2, the impact on cell proliferation and transactivation in glioblastoma and colon cancer cells, and molecular modeling to understand the interaction mechanism.

Regarding Figure 1, I suggest using a different template for illustrating bonds and atoms, because the resolution seems a bit low.

Additionally, while the 2D binding plots are informative, I recommend including a 3D representation of the binding interactions within the binding pocket for at least the most effective compound. This would provide a more comprehensive view of how the compound interacts with the receptor, potentially offering deeper insights into its mechanism of action.

For 2D plots, additional captions with amino acids are unnecessary, because they repeat information that is already in the figures.

Author Response

  • Figure 1 has been modified to improve the resolution
  • We have examined some of the 3D models (see below) and feel that the similarities and differences in ligand-binding site interactions were more informative in the 2D format.3D model

Reviewer 4 Report

Comments and Suggestions for Authors

This work by Prof. Stephen Safe and co-workers demonstrated the Bis-Indole based analogs as Dual Nuclear Receptor 4A1 (NR4A1) and NR4A2 Ligands (DIM-3,5 and DIM8-3,5). Activation of NR4A1- and NR4A2-dependent transactivation and cytotoxicity of these compounds colon and glioblastoma cells was also explained. Moreover, authors have also reported that these compounds inhibited tumor growth in a syngeneic mouse model of colon cancer using MC-38 cells as xenografts and analysis of T-cells in tumor infiltrating lymphocytes showed that the dual NR4A1/2 ligands reversed T-cell exhaustion. The manuscript can be accepted for publication after addressing the below mentioned minor comments.

1) The series of DIM-3,5 and DIM8-3,5 compounds are stated as positional isomers and differ only with respect to their substituents in fact these are not positional isomers. Authors should correct this.

2) In the materials and methods, in the NMR section, there are so many typos. 

a) Chemical shift ranges were reported for doublets and triplets. There should be sharp chemical shift value for doublets and triplets rather than the range.

b) Coupling constants were not mentioned for most of the doublets and triplets through out the data. Three coupling constant (J) values are mentioned for a dd which is supposed to have  only 2 J values. Two coupling constant (J) values are mentioned for a triplet which is supposed to have only 1 J value. Similarly for a doublet. Four coupling constants were mentioned for a quarteret, which will also have only 1 J value. Multiplets will not have a coupling constants at all (coupling constants were mentioned for multiplets in the data). For a broad singlet mention as bs (not s, b). Follow the standard protocols for writing the NMR data.

Entire NMR data has to be corrected and rewritten (so many errors throughout the data).

c) 13C NMRs should be reported

d) Mass spectral data should Also be reported.

e) If possible, provide the HPLC traces for most active compounds.

Author Response

  1. “Positional” has been deleted
  2. The typos have been corrected and the campus NMR service laboratory gave us ranges when the peaks could not be resolved. We have tried to minimize the errors based on the Reviewer’s comments and have extensively revised the presentation of this data.
  3. 13C NMR spectra were not determined and we did not measure detailed mass spectra but used it only to confirm purities and molecular weights.